theory of computing/e-science/bioinformatics

science of funding, science of success, collaborative funding, population health, science of science

**Author for correspondence:**
Kishore Vasan
e-mail: vasan.k@northeastern.edu

# The hidden influence of communities in collaborative funding of clinical science

Kishore Vasan[1] and Jevin D. West[2]

[1]Network Science Institute, Northeastern University, Boston, MA, USA
[2]Information School, University of Washington, Seattle, WA, USA

KV, 0000-0003-2610-1865; JDW, 0000-0002-4118-0322

Every year the National Institutes of Health allocates $10.7 billion (one-third of its funds) for clinical science research while the pharmaceutical companies spend $52.9 billion (90% of its annual budget). However, we know little about funder collaborations and the impact of collaboratively funded projects. As an initial effort towards this, we examine the co-funding network, where a funder represents a node and an edge signifies collaboration. Our core data include all papers that cite and receive citations by the Cochrane Database of Systemic Reviews, a prominent clinical review journal. We find that 65% of clinical papers have multiple funders and discover communities of funders that are formed by national boundaries and funding objectives. To quantify success in funding, we use a *g*-index metric that indicates efficiency of funders in supporting clinically relevant research. After controlling for authorship, we find that funders generally achieve higher success when collaborating than when solo-funding. We also find that as a funder, seeking multiple, direct connections with various disconnected funders may be more beneficial than being part of a densely interconnected network of co-funders. The results of this paper indicate that collaborations can potentially accelerate innovation, not only among authors but also funders.

## 1. Introduction

Clinical science studies are critical to advancing medicine and sustaining human health. They are testbeds for drug discovery and new medical devices. In coexistence with clinical trials, clinical science includes several stages of investigation to ensure the safety of test subjects, examining healthcare topics like effectiveness of drugs, diagnostic effects, epidemiological studies and much more [1]. They constitute an integral part of the clinical development process. It is estimated that the clinical evaluation process for a drug takes about 80.8 months from start to submission [2]. As a result, it is not surprising that funders spend almost $1 billion on

the development of a drug [3]. This enormous investment is fuelled by federal agencies, pharmaceutical companies and non-profit organizations along with several healthcare facilities and research institutes. During the rise of a new epidemic like COVID-19, funders, governmental and pharmaceutical alike, play a central role in allocating resources to speed up vaccine development [4,5]. We see the fruits of such quick mobilization in the case of the Johns Hopkins University with their $6.4 million seed grant at the onset of COVID-19 [6]; this grant which was then used to set up the National Convalescent Plasma Project to treat tens of thousands of patients [7]. In such a lethal global pandemic, it is important that funders work with collaborators to create new strategies for quick innovation [8].

It is estimated that the National Institutes of Health (NIH) consistently allocates one-third of its budget, a total of $10.7 billion, for clinical science funding [9]. On the other hand, the pharmaceutical companies in the USA are the primary driver of clinical research, accounting for 90% of total funding, which in 2015 would be an estimated investment of $52.9 billion.[1] It is also found that the bio-pharmaceutical industry invests as much as five times more in research and development, relative to their sales, than the average United States (US) manufacturing firm.[2] However, the research success for economic investment, and the benefit of building new collaborators from the perspective of funders, has been underexplored.

Grant funding is a cornerstone for the development and execution of a research project. While individual agencies can fund their own projects, multiple organizations can also collaborate with each other to fund similar projects of interest. As a clinical trial incurs huge investment with a low rate of success, sharing resources to reduce costs can be useful for funders, especially pharmaceuticals agencies, who are cost constrained [10]. Such collaboration can also expedite the clinical trial process by testing multiple therapies by multiple pharmaceutical companies on the same test group, relieving the stress of finding new participants [11]. Using this approach, there have been unprecedented achievements in identifying progression of Alzheimer's in the human brain which were made possible by data sharing between agencies [12]. There have also been trials that led to breakthroughs in understanding the biological system of Type 1 Diabetes patients, which was only made possible by a strong medical community of collaborators from industry, non-profits and academia [13].

More recently, collaborative funding led to the creation of the Accelerating COVID-19 Therapeutic Interventions and Vaccines initiative created by the NIH as a public-private partnership to prioritize drug candidates for trials and leverage assets among all partners.[3] Such collaborative methods have been argued as necessary for a sustained response in unprecedented times [14]. Even in non-clinical settings when collaborations need to be assembled quickly, connections between funders can be leveraged as it has been in the COVID-19 high performance computing consortium, which is used to provide supercomputers for researchers globally.[4] While there exists much anecdotal evidence citing the importance of collaborative funding, there is a knowledge gap in quantifying and measuring the impact of such collaborations. Mapping the landscape of collaborative funding will help identify strong and weak partners and eventually assist in building targeted science policies that promote research collaborations across authors, funders, institutions and countries.

## 1.1. Defining funder collaboration

In the broader context, inter-agency collaboration can happen in two ways.[5] First, two or more agencies may introduce a proposal call to jointly fund projects in a topic. For example, a grant call titled 'Clinical research for new therapeutic uses of already existing molecules (repurposing) in rare diseases', was jointly organized by 19 agencies [15]. It may also be possible that agencies may create initiatives that support a cause. For example in June 2019, the National Heart Lung Blood Institute of the NIH came together with the California Institute of Regenerative Medicine to co-fund cell and gene therapy programmes through the Cure Sickle Cell Initiative [16]. Another such example would be the creation of a joint venture, ViiV Healthcare, which was

---

[1]Pharmaceutical Research and Manufacturers of America (PhRMA) 2016 profile on bio-pharmaceutical research industry. Accessed from: http://phrmadocs.phrma.org/sites/default/files/pdf/biopharmaceutical-industry-profile.pdf. Pages 33–35.

[2]Congressional Budget Office (CBO) of the United States 2006 report on pharmaceutical industry. Accessed from: https://www.cbo.gov/sites/default/files/109thcongress-2005-2006/reports/10-02-drugr-d.pdf. Page 9.

[3]https://www.nih.gov/news-events/news-releases/nih-launch-public-privatepartnership-speed-covid-19-vaccine-treatment-options.

[4]https://covid19-hpc-consortium.org/.

[5]When referring to funding collaboration, we talk about both kinds of collaboration, unless noted explicitly. We include both types because it is out of the scope of this paper to delineate between these two kinds of collaborations at the scale of our study. One possible way is to identify the grants associated with publications and check if the funders were listed on the same grant. The current data are limited in such mappings and would require hand labelling hundreds of thousands of papers. This differentiation will be an important follow-up to this study.

founded by two pharmaceutical companies Pfizer and GlaxoSmithKline to fund HIV drugs [17]. More recently, a large coalition of philanthropic and private funders with the Coalition for Epidemic Preparedness Innovations has been pivotal in advancing vaccine development for COVID-19.[6] In such scenarios, multiple organizations agree to join forces to co-fund projects or therapeutics, thereby extending a formal agreement to collectively provide monetary support.

By contrast, multiple agencies may also be *unaware* that they are supporting the same project. Agencies can be acknowledged as funders through supporting the same co-authors. That is, multiple authors with different funding sources may choose to collaborate on a project or a single author may use multiple grants to support a study, thereby connecting any two agencies together via acknowledgement. While the two agencies acknowledged may not be aware of the collaboration link, they can still be regarded as connected since they are interested in funding similar topics and similar people. Such mechanisms of evaluating funding links from the perspective of author publications can enhance our understanding of the complexity of research funding realities [18]. This type of collaboration may be hidden to funders but it explains how authors collaborate, share ideas and receive support from similar funders. Therefore, the network of funders, where an edge indicates collaboration of a paper, signals the role of funders and authors in encouraging collaborative science. We suspect this form of unintentional co-funding as the commonest of the two types. For simplicity, we refer to both types as collaborative funding and the network thereof as a co-funding network.

In addition to informing us about collaborative choices, the co-funding network can also be viewed as a resource for social capital, one that an individual agency can use to its benefit. In [19], Burt describes social capital as the contribution of relationships, both within the organization and outside, to the success of an organization. It is a metaphor for social structure which signals that agents who are well connected in the network also perform better. The co-funding network is representative of a social structure because the connections between agencies amount to sharing of resources, ideas and economic goods towards a particular project. This represents a flexible collaboration environment that allows a funder to diversify its core skills to engage in exploration of new technologies that would have not been possible without this support network [20]. Using the co-funding network as a synonym for social interactions, we test the association of local connections to achieve higher success in funding. To measure success, we use a $g$-index type metric for agencies that measures the average citations of the top $g$ cited papers that are funded by an agency [21]. This is a useful metric because it allows us to standardize the citations to measure the research success of a set of papers funded by an agency. Because our data are of clinical science papers it allows us to measure the efficiency of a funder in funding clinically relevant research; but how do the collaborative ties of a funder influence its success in funding? Such a quantitative examination of the benefit for funders in seeking diverse partnerships and the contribution of collaborative funding in producing high impact research has not been done in the past and serves as a novel contribution of this work.

The primary goal of this paper is to examine the role of collaborations between funders and its implications in achieving higher research success from the perspective of funders. We conduct our research analysis in two broad steps. First, we map the connectivity of funders through a co-funding network and explore the characteristics of communities created through these collaborations. Next, we measure the research value generated through these funder collaborations. In particular, we control for authorship and compare the value of solo-funding versus co-funding in generating high impact research. Furthermore, we examine the network structures of funders that are associated with research success. This type of analysis may allow a funder to strategically position itself within this network. We also see this work as useful for researchers trying to identify connected funders and for policy makers examining the research success of different funding model combinations. While the underlying data pertain to clinical science, we believe that the methods used in this paper to examine funding connectivity can be applied to other scientific domains that depend critically on collaborative funding such as climate change and public health.

## 2. Background

Funders have a broad role in the development of science that includes the power to determine the next era of science research topics through rigorous grant proposal reviews [22] to picking the next researchers [23]. By virtue of providing financial support, funding agencies create the infrastructure required for scientists to pursue and conduct research, thus yielding a mediated and indirect influence on science

[6]https://cepi.net/covid-19.

[24]. While grant funding has minimal impact on the productivity of a researcher [25], it has been found that grant-sponsored projects get published in high impact journals [26] and therefore account for higher citations [27], a realization that persists across multiple domains [28,29]. It has been discussed that this is partly because a researcher with a funded project typically has access to better resources and funds to travel and is usually more motivated to conduct research than otherwise. The differences between funding and research outputs can also be affected by the grant mechanisms distributed to investigators (e.g. NIH's R01, R21) [30]. Overall, funders create the organizational structure for authors to conduct research, making them important participants of science. But how do we evaluate the success of funders? In particular, there is a lack of clarity on how collaborative ventures help funders produce higher research success which is the primary focus of this paper.

Previous work examining heterogeneous co-funding has found that projects funded through public-private partnerships is useful in establishing a stronger clinical research enterprise and that efforts to promote this should be better coordinated [31,32]. In that regard, countries like Canada have invested in promoting a co-funding model that encourages government-industry partnerships [33]. These connections between institutions are found to be governed by geographical proximity [34,35]. The role of physical distance is also found to be important in generating research and design collaborations among pharmaceutical companies [17]. Previous work argues that geographical closeness extends a shared trust enabling the alignment with funding similar areas of research, thus observing local collaborations at a higher rate than global collaborations. These papers provide us with the much needed background to interpret our findings of communities of funders driven by the country of operation.

The notion of social capital has been around for a long time and stems from the fields of economics, sociology, and social networks [36]. The premise is that agents (individuals, groups, organizations) can mobilize their network connections to attract high benefits. In many cases, social capital is intrinsic, and the reward of social interactions is the interaction itself [37]. In other words, agents seek to garner social capital as a means to use the combined resources of their partners. The ideas of pursuing social capital is motivated at an individual level, but it can also be useful at an organizational level, when thinking about resource dependence, market constraints and partnerships [38]. The social capital embedded within the firm-university collaborations is found to be crucial in facilitating knowledge transfer [39] and for building a common view of goals [40]. Moreover, institutions can overcome organizational boundaries by building a diverse range of alliances [41] and sustaining relationships by seeking local partners [42]. This previous work highlights the crucial value of networks at an organization level. For our project, the social network of funders is examined through the lens of publications, where multiple funders may be acknowledged together for supporting a project. We use this network to examine collaborative funding and its relationship to research success.

# 3. Methods

## 3.1. Data

The data used for analysis in this paper were obtained from (DATE), a Digital Science's Dimensions platform [43].[7] The Analytics API in Dimensions allows researchers to query publication metadata like authors, affiliations and funders from thousands of journals. From this database, we filtered all papers published in the Cochrane Database of Systemic Reviews journal, the largest producer of systemic reviews of clinical evidence that is read by clinicians worldwide.[8] The Cochrane journal is found to conduct a better assessment of methodological rigor compared to other journals [44]. We therefore refer to these papers as 'core papers' in clinical science and the citations within this journal to reflect clinically relevant publications. It is important to note here that because Cochrane is a European journal there may be citation bias present in the data (see Limitations and future work section).

We then extract the incoming citations (citing papers; papers that cite the Cochrane journal or forward citations) and outgoing citations (cited papers; papers that are cited by the Cochrane journal or backward citations). The citations within the Cochrane journal have been subject to multiple bibliometric analysis regarding journal [45] and citation searching [46] bias, along with examining the transfer of knowledge

---

[7]The propriety data are unavailable for open access. However, the data can be accessed at https://app.dimensions.ai with credentials.

[8]For ease, we will refer to the Cochrane Database of Systemic Reviews journal as 'Cochrane journal' and the papers from that journal as 'Cochrane papers'.

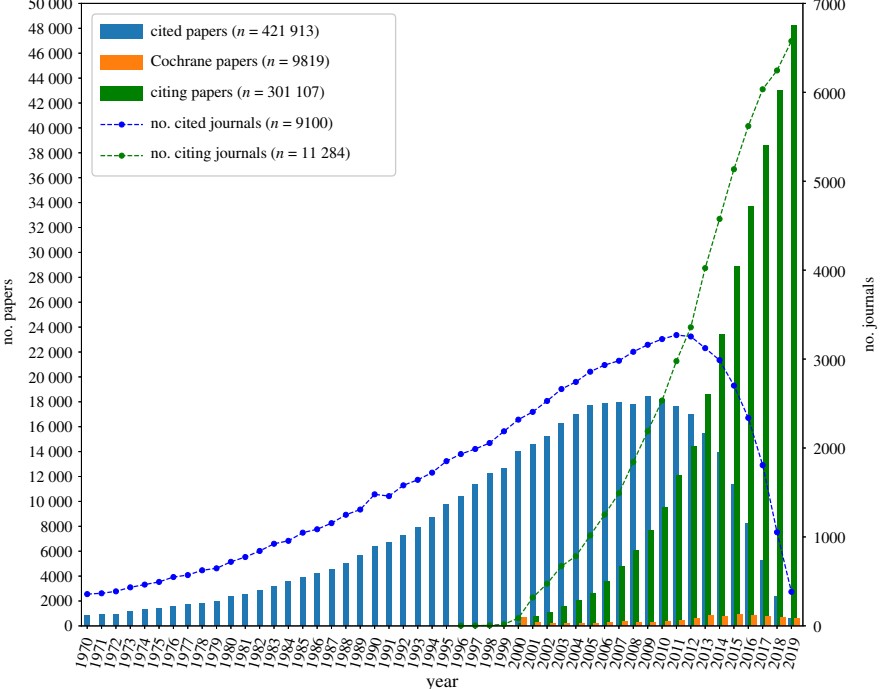

**Figure 1.** Paper and journal counts. Cited papers represent the papers that are cited by the Cochrane journal and the citing papers represent the papers that cite the Cochrane journal. We observe that a small subset of Cochrane papers cites and receives citations from a significantly larger list of papers and journals, indicating the wide readership of the Cochrane journal.

**Table 1.** Raw paper and journal counts of the core dataset from 1970 to 2019.

|  | no. papers | no. journals | papers w funders | no. funders | papers/ funder | papers/ journal |
|---|---|---|---|---|---|---|
| all papers | 713 514 | 14 036 | 159 063 | 1761 | 90.3 | 50 |
| cited papers[a] | 421 913 | 9100 | 87 647 | 1472 | 59.5 | 46.3 |
| citing papers[a] | 301 107 | 11 284 | 74 551 | 1645 | 45.3 | 26.6 |
| Cochrane papers | 9819 | 1 | 3844 | 378 | 10.1 | — |

[a]Does not include counts of Cochrane papers.

within systemic reviews to clinical practice [47]. We therefore consider the set of forward and backward citations to represent the larger sample of papers published regarding clinical science and therefore, combined with the 'core papers', represents the entire publication space, which serves as our 'core dataset' of 713 514 papers for the analysis (table 1). Owing to missing historical data, we restricted our analysis to papers published between 1970 and 2019 (figure 1). However, owing to limited representation of funding agencies prior to 1998, we subset our funding analysis to the papers published between 1998 and 2019 (see figure 2 for counts of funded papers over time and the electronic supplementary material for justification).

### 3.1.1. Funding data

Every paper in the Dimensions database contains metadata about the publication like the title, keywords, authors, research organizations, journal title and funding information. The funding information includes

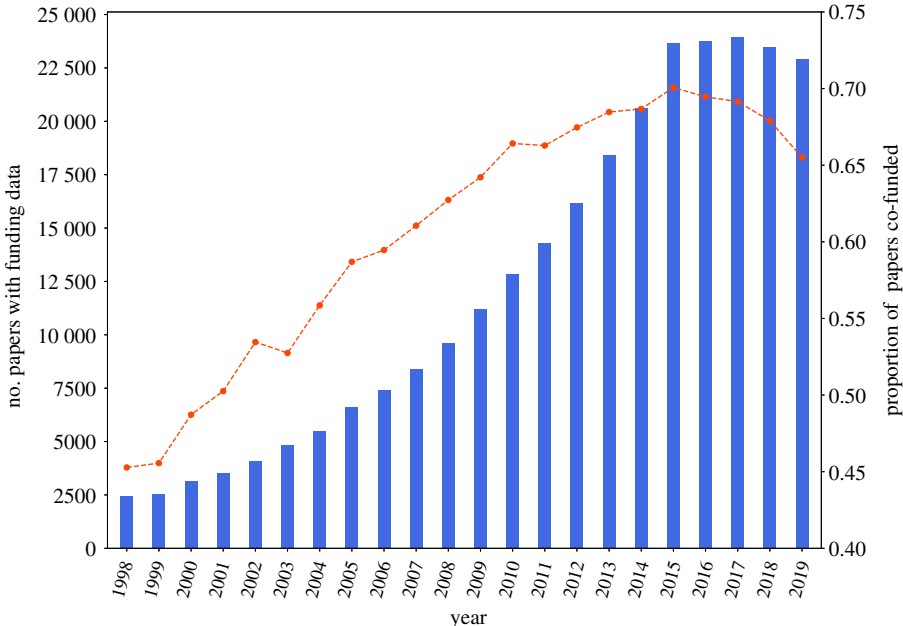

**Figure 2.** Proportion of co-funding over time. We observe that the proportion of papers co-funded (line) has increased over time, indicating that majority of the publications have support from multiple funders.

the name of the funder along with the city, state, country and type of organization, and the grants associated with the papers. A funding organization is acknowledged in a paper if it provided support, monetary or otherwise, that made the work possible. In our analysis, we do not differentiate between monetary support and non-financial support; therefore, the relationship between funders encompasses collaboration and funding. We present the proportion of papers funded by top 20 funders in figure 3.

### 3.1.2. Types of funders

The funders are further sub-categorized as ones that are government (e.g. National Institutes of Health Research (NIHR)), company (e.g. Pfizer), non-profit (e.g. Bill and Melinda Gates Foundation), facility (e.g. Dana-Farber Cancer Institute), healthcare (e.g. Chang Gung Memorial Hospital), archive (e.g. US National Library of Medicine), education (e.g. King's College London) and other (e.g. São Paulo Research Foundation). We apply these categories to an analysis of collaboration patterns between multiple types of funders. Dimensions standardized and indexed agency names so there was limited manual inspection necessary.

## 3.2. Co-funding network

We explore the relationship between funding agencies by creating a weighted network of agencies where nodes are agencies, and a weighted link indicates the strength collaboration. From the list of funders, we remove funders who have published less than 25 papers (see the electronic supplementary material). This threshold yields a representative sample of 653 funders and 143 904 papers. Out of these papers, 84 613 papers are funded by a single agency while 59 291 papers are co-funded. We use this list of funders and papers to create the co-funding network. This gives us a network of 653 nodes and 26 165 edges, with an average degree of 80.13.[9]

### 3.2.1. Communities in funding

To find communities of funding agencies, we use the Louvain community detection algorithm [48], which decomposes the network in an unsupervised fashion into sub-units or modules that are highly

---

[9]The code to recreate the analysis, along with the network data will be made publicly available.

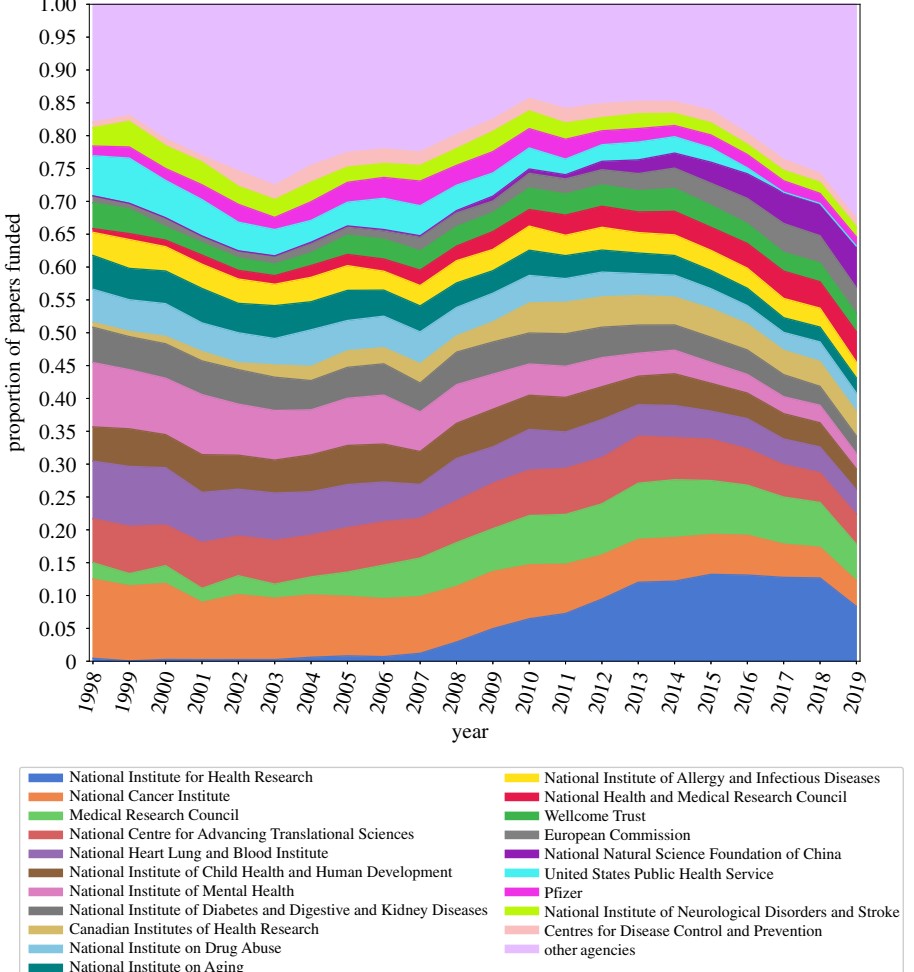

**Figure 3.** Proportion of papers funded by the top 20 funding agencies. We find that most agencies remain constant in the number of funded projects over time. Additionally, we observe that National Institutes of Health Research has increased its contribution to clinical research, about 10% annually in 10 years after being founded in 2006, while the contribution by the National Cancer Institute has decreased over the past 20 years. We also find that the top 20 agencies collectively fund about 80% of the papers every year. The drop off in proportions over the past five years could be owing to the uptake in new funders.

interconnected (see the electronic supplementary material for more detail). A visualization of the co-funding network is presented in figure 4.

### 3.2.2. Heterophily in funding

A key driving feature in the performance and success of an organization in the business world relates to the diversity in social connections. By analogy, collaborations between different types of funders (e.g. pharmaceutical and federal) might facilitate the influx of new ideas about and approaches towards problems in biomedicine. The network of funders allows us to conduct such analysis about preference of heterogeneous network ties (e.g. the NIH collaborating more with non-profits than with pharmaceutical companies). To evaluate diversity in collaborative funding, we use the classification of the funders present in the Dimensions data as node types. This gives us a way to quantify the diversity in network ties from −1 to 1, with −1 being least diverse and +1 being most diverse. The heterophily score accounts for the number of dissimilar connections ($E$) and the number of similar connections ($I$) of a funder with its neighbours to create a $EI$-index for heterophily [49]. The equation for this is described in equation (3.1). We present the distribution of $EI$ scores for different types of funders in figure 5 and the number of edges in table 2:

$$EI_{\text{index}} = \frac{E - I}{E + I}. \tag{3.1}$$

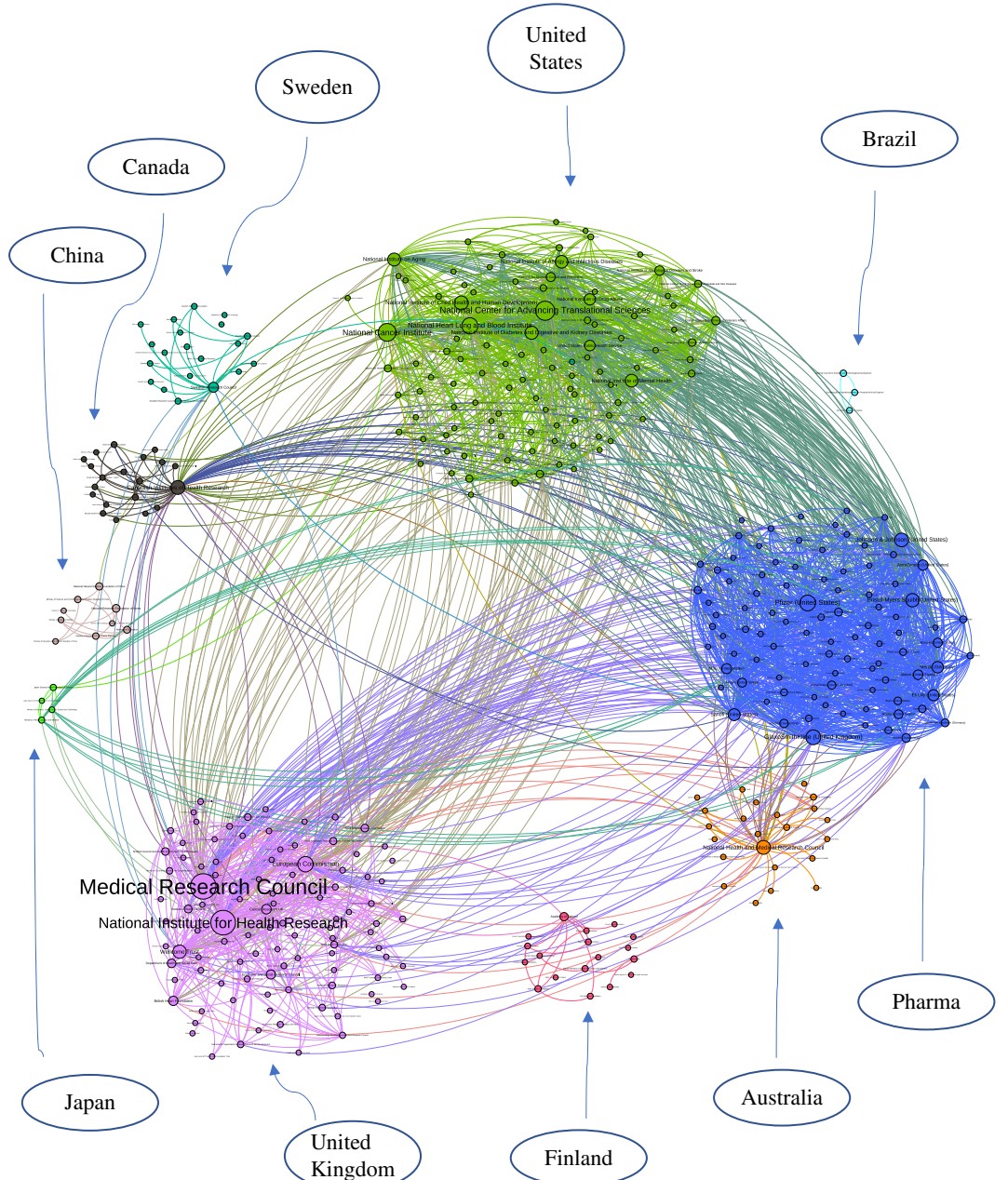

**Figure 4.** The co-funding network. We observe 10 different communities that are guided by the primary region of operation and the primary goal of the agency, suggesting that co-funding efforts between funders take place at a country level. Additionally, we also observe a large community of pharmaceutical communities collaborating together, pointing to strong resource sharing among pharmaceutical companies for the conduct clinical science research for a particular drug or product. Nodes are sized by PageRank scores.

## 3.3. Measuring agency success

Author reputation and journal prestige can influence citations, irrespective of funder impact. This makes it difficult to estimate the role of a funder in that paper. We need more than just raw citations. We need to compare big funders that may support several low-impact publications, and small funders that may have a few high impact papers. Ultimately, we want to compare entire funding portfolios. There is no perfect measure for this, but we resort to a metric that gets closer to this ability than just comparing citation counts. The metric is called a $g$-index. Mathematically, the formula finds the highest possible $g$ using this equation:

$$g <= \frac{1}{g}\sum_{i<=g} c_i. \tag{3.2}$$

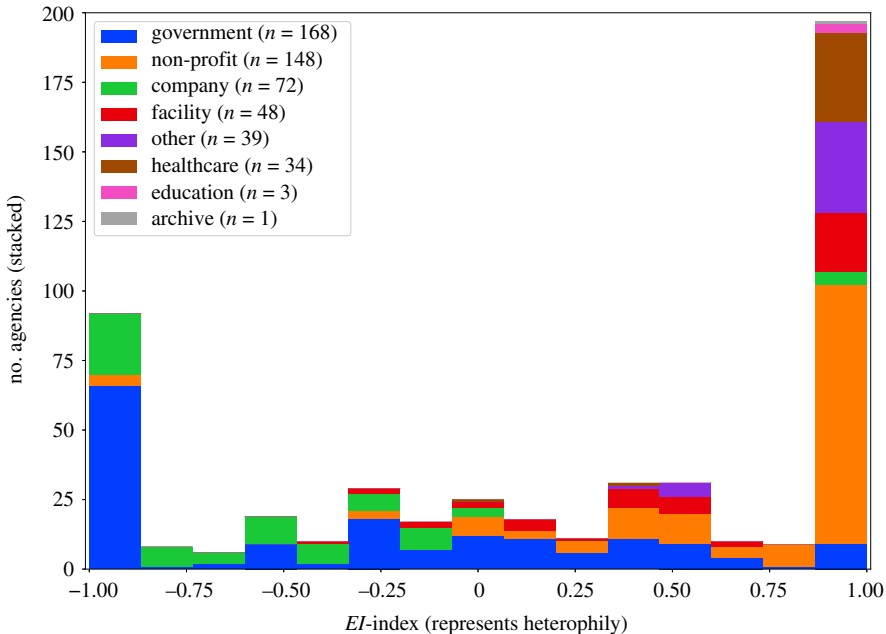

**Figure 5.** *EI*-index to quantify diversity in co-funding, where agencies towards −1 show limited diversity in social ties while agencies towards +1 show high diversity. We observe that most governmental organizations and pharmaceutical companies are less diverse, indicating that these funders collaborate with similar funders. On the other hand, the majority of non-profit organizations are very diverse in their co-funding choices. Other funders persist across the scale, although most of them show high diversity in their choice of co-funder.

**Table 2.** Number of edges between different types of funders. (One of the key observations here is that companies collaborate mostly with other companies and that non-profits funders collaborate mostly with government funders.)

|                  | (1)  | (2)  | (3) | (4)  | (5) | (6) | (7) | (8) | total             |
|------------------|------|------|-----|------|-----|-----|-----|-----|-------------------|
| (1) government   | 834  | 325  | 394 | 360  | 74  | 5   | 5   | 73  | 2070              |
| (2) company      | 325  | 1478 | 149 | 220  | 13  | 0   | 0   | 10  | 2195              |
| (3) non-profit   | 394  | 149  | 142 | 164  | 22  | 0   | 1   | 27  | 899               |
| (4) facility     | 360  | 220  | 164 | 316  | 29  | 8   | 0   | 39  | 1136              |
| (5) healthcare   | 74   | 13   | 22  | 29   | 2   | —   | 1   | 2   | 143               |
| (6) archive      | 5    | 0    | 0   | 8    | 0   | 0   | 0   | 1   | 14                |
| (7) education    | 5    | 0    | 1   | 0    | 1   | 0   | 0   | 1   | 7                 |
| (8) other        | 73   | 10   | 27  | 39   | 2   | 0   | 0   | 8   | 160               |
| total            | 2070 | 2195 | 899 | 1136 | 143 | 14  | 7   | 160 | 6624[a]           |

[a]3312 unique pairs of edges.

That is, an agency will have a $g$-index score of $g$ if the average number of citations of the top $g$ cited papers is at least $g$ [21]. In other words, the $g$-index sorts papers in descending order based on number of citations and finds the highest number of $g$ articles that together received $g^2$ citations. Therefore, this metric accounts for the number of papers funded along with the quality of those papers (citations), both of which are indicators of success to a funder.

We chose this metric over others available because it better captures the highly cited papers. An alternative metric, such as the well-established $h$-index, does not reward highly cited papers. As a consequence, a few highly cited papers alone are not enough for a high $h$-index. For example, if one assumes a citation distribution where one paper has 1 k citations, and the other 99 have zero, the funder would have an $h$-index of 1 and a $g$-index of 31. By preferring a $g$-index over an $h$-index, we are able to give higher rewards to funders for identifying impactful avenues of research, an important pursuit in science funding. Finally, this metric allows us to compare funders with different funding

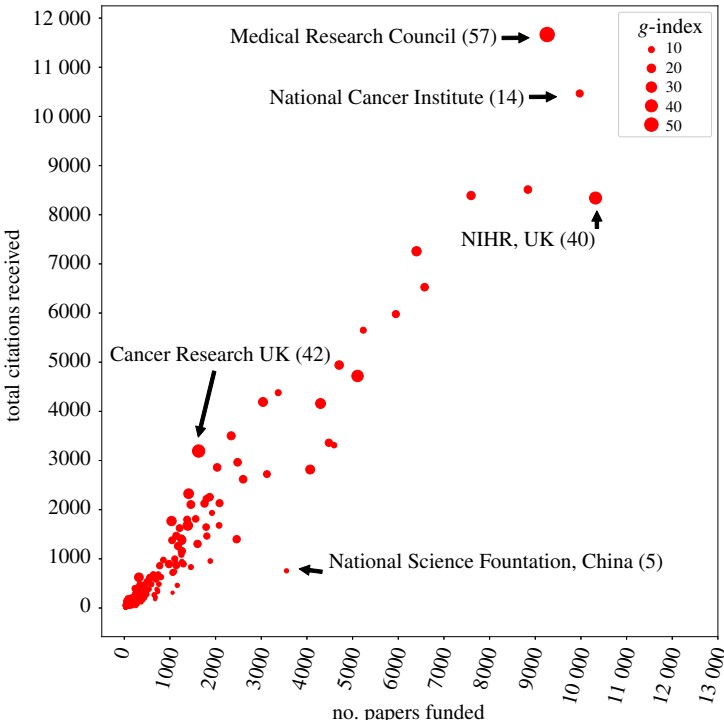

**Figure 6.** *g*-index description. We see that the *g*-index metric is able to easily contrast the number of papers funded, which usually depends on the funding capability of the agency, and the number of citations received, which is usually dependent on the journal. Using this measure, we are able to identify outliers like Cancer Research UK which has a higher score than NIHR and the National Cancer Institute, despite funding much fewer papers.

models and priorities. For example, we can now compare a funder with 10 $100 k grants and a funder with 100 $10 k grants, where the latter funder is expected to produce more papers. We present detailed examples of how this metric considers the number of papers and number of citations when calculating the *g*-index score in figure 6.

Although this measure can act as a generalized score for evaluating agency performance, the current analysis focuses on the citations received by non-Cochrane papers from Cochrane reviews, thereby removing journal self-citations. The reasoning here is that if a funder funds projects that are in related clinical science journals like the British Medical Journal (BMJ), receiving a citation from a Cochrane review is an indication that the published paper in the BMJ is clinically relevant. Therefore, looking at the number of citations received by the BMJ paper helps measure the clinical relevance of that paper and thereby indicates the performance of a funder in funding clinically successful projects. We will use only the citations received from Cochrane papers for subsequent citation analysis. We present the top 20 ranked agencies based on this metric in table 3.

### 3.3.1. Co-funding versus solo-funding

To compare the difference in success of collaborative funding and solo-funding we first controlled for authorship (which is known to drive higher citations [50]) by sub-classifying papers as written by a single or multiple authors. Of the single-authored papers, 2660 (70%) are solo-funded while 1162 are co-funded (30%). Of the multi-authored papers, 83 689 (61%) are solo-funded while 52 576 are co-funded (39%). This classification then gives us 207 funders who solo-fund and co-fund among single-authored papers and 647 funders who solo-fund and co-fund in multi-authored papers. This allows us compare the success of a funder across collaborative funding and solo-funding. We present the g-index comparison for each type of funding choice in figure 7.

### 3.4. Ego network in co-funding

The global network structure and its characteristics (e.g. small world, scale-free) provides key insights into the efficiency in information travel and its robustness to failure. Similarly, the local structure, that

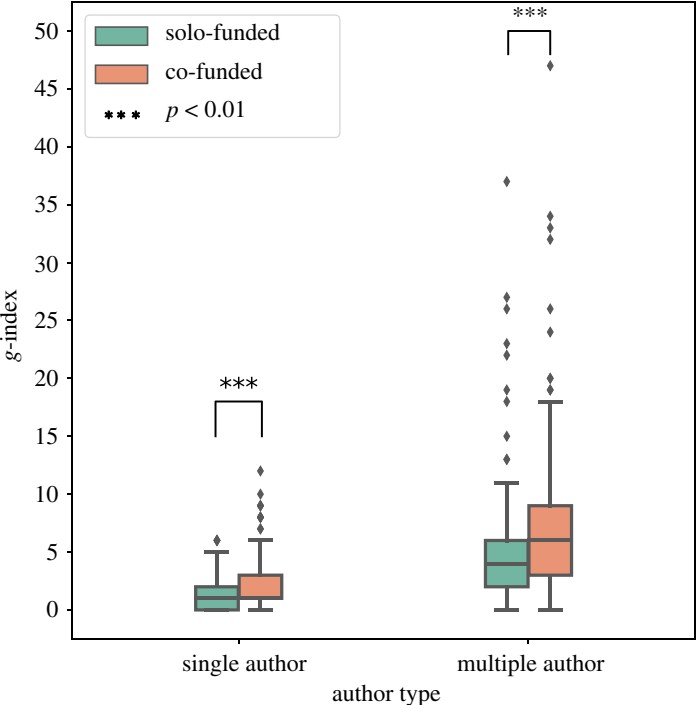

**Figure 7.** Comparing solo-funding and co-funding. We see that even after controlling for single authors and multi-authors papers, an agency achieves higher success through collaborative funding than solo-funding. This shows that collaboration is an inherently useful mechanism at the author level and at the funder level.

**Table 3.** Top 20 ranked agencies using *g*-index measure. (Note: National Institutes of Health is separated into different institutes.)

|  | agency name | *g*-index |
|---|---|---|
| 1 | Medical Research Council | 57 |
| 2 | Cancer Research UK | 42 |
| 3 | National Institute for Health Research | 40 |
| 4 | Canadian Institutes of Health Research | 36 |
| 5 | Wellcome Trust | 27 |
| 6 | Merck | 26 |
| 7 | NIHR Evaluation Trials and Coordinating Centre | 25 |
| 8 | Abott | 24 |
| 9 | National Institute of Mental Health | 24 |
| 10 | European Commission | 22 |
| 11 | Pfizer | 22 |
| 12 | United States National Library of Medicine | 21 |
| 13 | National Institute on Aging | 20 |
| 14 | National Heart Lung and Blood Institute | 19 |
| 15 | British Heart Foundation | 18 |
| 16 | GlaxoSmithKline | 18 |
| 17 | Bristol-Myers Squibb | 17 |
| 18 | Novartis | 16 |
| 19 | Centers for Disease Control and Prevention | 16 |
| 20 | AstraZeneca | 16 |

**Table 4.** Correlation values (Spearmann). (We compare the association between different network structure metrics and the *g*-index success metric. Numbers in italics signify values mentioned in the Results.)

| | (1) | (2) | (3) | (4) | (5) | (6) | (7) | (8) |
|---|---|---|---|---|---|---|---|---|
| (1) *g*-index | 1.00 | — | — | — | — | — | — | — |
| (2) deg centrality | *0.71* | 1.00 | — | — | — | — | — | — |
| (3) bet centrality | *0.51* | 0.67 | 1.00 | — | — | — | — | — |
| (4) density | *−0.60* | −0.75 | −0.97 | 1.00 | — | — | — | — |
| (5) constraint | *−0.67* | −0.92 | −0.61 | 0.68 | 1.00 | — | — | — |
| (6) effective size | 0.72 | 0.97 | 0.73 | −0.80 | −0.94 | 1.00 | — | — |
| (7) efficiency | −0.34 | −0.65 | −0.07 | 0.09 | 0.35 | −0.16 | 1.00 | — |
| (8) frag index | −0.31 | −0.63 | −0.19 | 0.21 | 0.54 | −0.54 | 0.84 | 1.00 |

is the node of interest (ego) and its neighbours (alters), has important hidden characteristics that allow the node to succeed [51]. This form of evaluation is useful because it exhibits the information available to the ego and the social influence of its alters. Consistent with previous social science and network science literature, we associate the structure of the ego network of the nodes to its success. Such analysis allows us to infer about network connections and its success.

In our analysis, the ego network of a funder helps it navigate social ties to create future collaborations with other funders. New funder collaborations require serious discussion concerning governance structures, data management and project timelines [52]. Indeed, building successful research projects require good communication and a sense of collective vision among the funders. Such collaborations are often tough to create and sustain owing to varied scientific priorities, particularly around the disbursement and use of funds [53]. These differences are further reinforced because of national funding policies and rigid disciplinary boundaries, making certain funders 'distant' and unapproachable. Such collaborations can be effectively mediated by using the network of funders and it is possible for funders to ease the constraints of collaborations through policy changes.

However, the benefits of seeking distant partnerships is unclear. To test this effect, we use the *g*-index measure described in the earlier section as an output variable to examine the association of network connectivity and research success. The different network measures may indicate the effectiveness of resource sharing as well as the role of the ego in enabling efficient flow of information (see the electronic supplementary material). In other words, these network measures identify the role played by funders in building varied partnerships and establishing itself as a key funder in clinical science. As noted earlier, it can be expensive to create new connections. Funders therefore may more often fund within their internal network in trying to maximize their investment. We want to test the pay-off of distant connections in the network with the following hypothesis.

### 3.4.1. Hypothesis 1 (H1): co-funding with multiple, disconnected funders is negatively correlated with research success

For our analysis, we consider an edge to be significant if the two agencies have collaborated on more than 10 papers together. This eliminates collaborative efforts that are less meaningful and reduces other factors, e.g. a one-time author collaboration that could drive the network. Beyond this threshold, we find a network with 512 nodes and 3312 edges with an average degree of 12.93 from 53 654 papers with 127 079 (20% of overall) citations from Cochrane papers. This reduced network provides a strong indicator of collaborative decisions between funders. We present the correlation values of the network measures and the *g*-index success metric in table 4.

## 4. Results

### 4.1. Communities of funders

The co-funding network created using funding acknowledgement in publications, while the edges may be implicit to funders, provides important information about how money flows in science between

authors, institutions, funders. Through this constructed network, we discover communities that are inspired by the geographical location and the primary funding objective of the funders (figure 4). We find 10 different communities, with a maximum modularity score of 0.48, nine which consist of organizations that are co-located in the same country or region. This form of division in co-funding choices points to the localization of funding disbursement. In addition to large communities of funders in the USA and UK, we also see smaller communities of funders in Sweden, China, Japan and Brazil that are highly interconnected with each other. Probably, these communities could arise organically owing to the proximity of funding organizations and/or researchers who know of and work with one another [35]. As noted in previous work, a reason for geometric proximity between funders could be explained either because it allows for easy communication or higher visibility for actions, or even a shared regional culture.

Overall, we find that funders are more likely to collaborate with others in the same country than outside the country. Of all the papers that had multiple funders, 39 517 (73.5%) publications had funders from the same country while only 14 221 (26.4%) publications featured funders from multiple countries. Indeed this observation could be attributed to the funding policies from national entities that focus on regional collaborations, thus creating interdisciplinary boundaries between countries [54]. However, if such restrictions did not exist, how would the collaborative funding landscape look? Based on random simulations, we find that if funders pursued collaborations with funders irrespective of their country, 57.70% of the co-funded papers would feature international collaborations.[10] This indicates that the real-world network has much fewer international partnerships than we would expect at random. In other words, the country of operation highly influences the ability for two funder to collaboratively fund a project. Our findings offer further confirmation to the hypothesis of localized science, where resource sharing and exchange of ideas appears to be country mediated.

In addition to localized collaborators, our analysis reveals a tightly knit community of pharmaceutical companies. These funders tend to have both a global presence and a shared interest in the type of projects they fund. Although companies like Pfizer, GlaxoSmithKline, and Johnson and Johnson have headquarters in different countries, they still collaborate and share resources with each other for projects that conduct similar research. We acknowledge that this finding appears to be a contradiction to the commonly held notion that pharmaceutical companies are competitive and protective of their clinical programmes. Stepping back however, there are several factors that would seem to prompt collaboration between companies. First, there is a clear value to sharing of resources, especially in setting requiring testing at multiple hospitals and research institutes [10]. This makes it essential for some pharmaceutical companies to actively collaborate with others. Second, it is possible that the pharmaceutical companies who collaborate with each other often have the same parent company thereby relaxing concerns about intellectual property concerns and easing the co-funding process. In addition, pharmaceutical companies often have a different outlook to funding than do other governmental organizations. The latter organizations (e.g. the NIH) are known to fund basic science related to biological targets for drug action while the former fund drug development [55]. This points to the demarcation of interests of the funding objectives of various agencies, possibly explaining the collaborative interest among pharmaceutical companies even though they are not co-located in the same region.

## 4.2. Networks of successful funding

The focus of success from the perspective of funders provides us a new lens into evaluating the value of current funding mechanisms. Using the *g*-index metric to evaluate citations, Cancer Research UK (*g*-index 42) is revealed as a higher-ranked organization, despite the fact that it funds fewer papers and has a smaller number of citations than for example, the National Institutes of Health Research (*g*-index 40) (figure 6). Reciprocally, this approach identifies funders of clinical science that are unable to achieve a higher score, like the National Cancer Institute (NCI) (*g*-index 14), despite funding more papers than the Medical Research Council (*g*-index 57). We observe such a pattern because the success metric devised here examines the citation distribution of all funded projects. This is why although NCI has funded many papers and has received many citations, it has a low number of highly cited papers, which reduces its success score.[11]

---

[10]To create random realizations, we conduct double edge swap between funders in the network, that is we would swap two funders between their funded publication. We draw the results from 10 randomized networks and take the average of the findings.

[11]We note limitations in this finding owing to citation bias that is present in the data (see the Limitations and future work section).

Next, we tested the role of collaborations in achieving higher success. This leads to one of the major findings of the study in that co-funded papers tend to be associated with a higher number of citations, after controlling for authorship (figure 7). We find that within the single-authored papers, the average $g$-index is 1.18, while it is 2.01 for co-funded papers (Welch $t$-test stat: $-4.71$, $p$-value: $1.5 \times 10^{-6}$). This difference is even more marked in the context of papers authored by multiple investigators where co-funding leads to an average $g$-index of 7.4 while solo-funding leads to an average of 4.9 (Welch $t$-test stat: $-4.73$, $p$-value: $1.7 \times 10^{-6}$). Across single-authored and multiple-authored papers, we find that funder success is higher for collaboratively funded papers. This type of formalization from the perspective of the funder is useful in realizing policy initiatives that promotes collaborative ties between funders.

As we noted earlier, the ego network of a funder may be a key driver in its success. To test the effect of network connections, we investigated if seeking diverse collaborations is negatively correlated with research success (table 4). We find that as a funder, having multiple different connections (degree centrality) and mediating those connections (betweenness centrality) has positive correlation (0.71 and 0.51) with the $g$-index score. This shows that retaining a central position in the ego network allows for an agency to accumulate multiple exchanges with multiple funders, which then boosts its social capital.

Conversely, being involved in dense ego networks (high density of interconnections) and highly constrained ego networks (where there are multiple connections among the alters) is negatively correlated ($-0.60$ and $-0.67$) to agency success. The relationship of high constraint and low success is well realized in multiple domains [56,57]. This is because, in a highly constrained network, the ego node is limited in access to information that spans diverse alters, giving less opportunity to recombine information and thereby fails to produce high impact. Using these results, we can reject our original hypothesis that collaborations between various different funders, in addition to being tedious to establish, is not beneficial. Rather, our analysis suggests that funders, instead of focusing attention on a select few partners, may consider to collaborate with multiple organizations that are disconnected among themselves. Such a ego network structure of a funder can be naturally formed by diversifying its funding portfolio of investigators, institutions and collaborators. While these results do not exhibit a causal relationship to success, we see that such a collaborative structure is associated with efficient information flow and resource sharing, leading to higher social capital and potentially higher return on research investment.

## 5. Discussion

The priority of every funder is to efficiently allocate resources to fund rigorous investigators or groups that would achieve the biggest scientific, societal and monetary impact. This gives funders an important role in accelerating science by means of funding research that may have high impact. A mechanism to promote high impact research is through collaborations. As we argue in this paper, the reward of collaborative funding is not merely individual highly cited projects, but also the intrinsic transfer of social capital between funders. Indeed, collaborations between funders on a paper, although hidden, may serve to foster the development of future partnerships. Such a dedicated effort to find collaborative opportunities will help create a centralized resource for collective knowledge which can be useful for science. We can see the application of the network of funders in helping researchers with the critical data sharing infrastructure for Alzheimers disease research.[12] While it is true that the tacit knowledge exchange from publications is often among authors, funders may find these hidden connections useful in funding projects when resources are limited. In a bottom-up view, the social connections formed through a publication can be beneficial to authors and funders alike.

We note that the drivers of success in collaborative funding could be confounded with other factors. It is possible that a research project receives funding from multiple funders owing to the inherent quality of the research or the reputation of the authors. On the other hand, funders may be acknowledged for infrastructure investment. It is indeed difficult to conduct interventions to truly tease out the value of funders in a project and by extension the value of their collaborations. However, the patterns of connectivity between funders can be useful to both academics and policy makers. As we find in this work, funders regionally co-located are much more likely to share resources than otherwise. This is a useful finding for authors and academic investigators as they begin to build their own personal network of funders for their research. Evaluation of science policies is an important component of

---

[12]Alzheimer's Disease Data Initiative (ADDI). https://www.alzheimersdata.org/.

research funding. In this work we find that funders self-organize into national groups and funders that have higher success tend to occupy a central position collaborating with multiple disconnected funders. We view this work as a blueprint for policy makers and science managers to make inferences about funding policies that enable knowledge spillover across countries and examine the research output of such collaborations.

In summary, the findings of this paper underscore and provide evidence for the intuitive belief that collaboration—be it between authors, departments, institutions or funders [58,59]—represents a major driving force in science. An underlying motivation for such collaboration, be it for authors or for funders, might be attributed to incentives associated with success. Funders have a high incentive to appear collaborative if authors continue to collaborate as well. At one level, the rise in collaborative funding can be attributed to an increasing embrace of multidisciplinary 'team science' [60]. At a more granular level, collaborations between funders can also galvanize multidisciplinary perspective and insight. Thus, the existence of links between two disease specific funders (e.g. the NCI and American Heart Association) reveals collaboration between investigators specializing in unlinked fields, leading to opportunities for cross-fertilization and the emergence of novel concepts, a well-documented feature of a paper with a high number of citations [61].

Furthermore, collaborative funding, as we have described here, accordingly, can play a forcing function for cross-disciplinary collaborative opportunities between authors and for science advancement. The presence of communities of funders is not insular but rather inter-weaved with the growth of science in recognizing the value of collaboration at different levels. As the scale and complexity of the problem increases, new models of collaboration will be essential to drive impact. We look at the network of funders as another dimension that explains how authors collaborate, share, and exchange ideas, and as an apparatus that can be modified with policy decisions to improve science. The network dynamics discussed here are not restrictive to funders; rather, they offer an approach to accelerate innovation in the scientific ecosystem, one that incorporates the role of funders in the complex web of ties between authors, institutions and fields.

# 6. Limitations and future work

As with every other big data related project, this one is not immune to limitations. Although the Cochrane Database of Systemic Reviews provides a unique overview of those papers published in the clinical sciences, it is not a verified benchmark dataset. Because the Cochrane journal is a European journal, it is possible that the papers which cite and get cited are weighted towards European journals read by European authors. As a result, we may observe bias in clinical science funding from European funders who generally fund European authors who then publish in European journals. It is also important to note that the funding information is subject to the data collection, extraction, and curation techniques employed by the Dimensions team. As of June 2021, the database is estimated to match 101 k organizations from 219 countries. This feature is well-integrated with the Digital Science portfolio of companies. The Dimensions API is constantly curated and matched with the up-to-date identification of organizations.[13] Furthermore, it is important to note that funding information included in only about 30% of the total papers published in any year has funding information (see the electronic supplementary material). This could either be owing to data collection issues or the convention of the authors/journals to not acknowledge any funders.

## 6.1. Endogeneity problem

A major drawback of our current work is the value of endogenous variables that may influence a high impact work. For example, a publication may receive funding owing to the inherent quality of work or the presence of established researchers and as a result attract more citations. In this case, the value of funder collaborations is rather limited and confounded with other parameters that produce high impact research. Our current research set-up prevents us from conducting interventions at the time of publication to test the value of funders. In particular, would the publication have still received a high number of citations if it did not receive support from multiple funders? This is an important question but through our results we are unable to make prescriptive claims about where to allocate funds and

---

[13]More information can be found at https://www.dimensions.ai/blog/boost-your-research-organizational-data-using-grid-and-the-dimensions-api.

who to collaborate with. We thus refrain from making causal claims about collaborations but instead present correlated associations between funder partnerships and research success. Furthermore, in the current analysis the investment of funders is unknown. For instance, the 'role' of the funder in a study (i.e. research design, resource sharing, monetary support) also affects the involvement of the funder and in translation their investment in seeking new collaborations. The current data do not allow for such fine-grained examination but is presented as an interesting avenue for future work.

# 7. Conclusion

Funding decisions and collaborative funding choices are very crucial to the success of biomedical research. When funding a project, a funder accounts for the cost and the estimated reward of supporting it. In this work, we explored the hidden effect of social connections formed by funders and its impact on research success. Using the $g$-index metric as a proxy for success we find that collaboratively funded projects are associated with higher success, both in the setting of single- and multi-authored papers. The network dynamics between funders indicate that collaborations between multiple disconnected funders can lead to higher impact than those between a few highly connected funders. We also find evidence that the position of a funder in the collaboration network can have hidden influence on the research success for economic investment. While not clearly a causal relationship, our analysis suggests that collaborative opportunities enable the efficient exchange of resources and ideas, leading to higher social capital and thus is associated with higher funding success.

Finally, we observe distinct silos of funders with a higher level of collaboration based on their primary country of operation, revealing a localized model of funding disbursement mediated at a regional level. We find that funders are more likely to collaborate with other funders within the same country, at a much higher rate than expected at random. This indicates the strong influence of geography in creating funder collaborations. On the other hand, we discover a tightly knit community of pharmaceutical companies, despite the fact that they are headquartered in different countries, suggesting an openness to resource sharing so that shared objectives can be achieved. We look at these findings as an amalgamation of how authors, institutions, and funders decide to work on projects collaboratively. The network analysed here presents another outlook for funders and policy makers to re-imagine a funding coalition that accelerates scientific innovation and promotes collaboration.

Data accessibility. The network data of funders along with their g-index scores, number of papers, number of citations, community information is publicly released via Dyrad: 10.5061/dryad.2jm63xsnq [62]. The entire publication corpus can be downloaded from https://app.dimensions.ai. The code to recreate the results using the provided data is available at: https://github.com/kishorevasan/clinical-funding.

Authors' contributions. K.V. and J.W. conceived the research. K.V. conducted the analysis (the majority of the research was conducted while at the University of Washington). Both authors wrote and approved the manuscript.

Competing interests. The funders had no role in the design and experiments of the study.

Funding. This work was possible because of funding support from the Bill & Melinda Gates Foundation, Mary Gates research scholarship; and travel support from the Mary Gates Endowment, Paul Jones Endowment, and the Population Health Initiative at the University of Washington.

Acknowledgements. The authors would like to thank Digital Science for providing the Dimensions data used in this research. We are indebted to Joseph 'Mike' McCune and Brian Doehle of the Bill & Melinda Gates Foundation and Carl Bergstrom of the University of Washington for the useful discussions. We also thank Alexander Gates of the Network Science Institute for assistance with the Dimensions data. We are grateful to the reviewers for providing critical feedback and helping us improve the manuscript.

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
