## [Peer Review File · Royal Society Open Science]

Review History

RSOS-210072.R0 (Original submission)

Review form: Reviewer 1

Is the manuscript scientifically sound in its present form?

No

Are the interpretations and conclusions justified by the results?

Yes

Is the language acceptable?

Yes

Do you have any ethical concerns with this paper?

No

Have you any concerns about statistical analyses in this paper?

No

Recommendation?

Major revision is needed (please make suggestions in comments)

Comments to the Author(s)

Studying the influence of research funding is important to a wide and multidisciplinary academic audience but also for science managers and politicians. While I think that the overall research topic is suitable for the audience of Royal Society Open Science, I also think that the manuscript needs substantial improvements. Below I provide a more detailed discussion of some questions and comments that I have. I encourage you to thoroughly address the issues outlined below to improve your manuscript.

Main Comments

1. Research Question and Contribution

After reading your manuscript multiple times, I am still not sure what the research question is. It seems as if different versions of the research question come up throughout several sections of the manuscript, e.g. “how inter-connected are the agencies?” (page 1), “But how does collaboration between funders take place and how does it affect research success?” (page 2) “we quantify the importance and influence of collaborative funding in clinical science research from the perspective of funders” (page 4), or “how do the collaborative ties of a funder influence its success in funding?” (page 4). All these different versions indicate different foci of the paper and the analysis. I strongly recommend using one concise research question throughout the manuscript that is strongly linked to our empirical strategy. This will also support you in being clearer about your contribution.

2. Related Literature – Social Capital

You refer to the concept of social capital as a key reference point in the existing literature. However, I am not yet fully convinced that this concept can be applied in the context of your study. Burt and other authors refer to social capital as an individual level concept. However, you are essentially analyzing links among organizations. Hence, it needs to be thoroughly discussed whether and to what extent the concept of social capital can be applied on the organizational level. Moreover, I do not see the merits of this concept for guiding the reader through the paper and the analysis. The focus of the analysis does not seem to be on examining the role of social capital. Instead of focusing on social capital I recommend elaborating in much more detail on the related literature on research funding (e.g., Braun, D. (1998). The role of funding agencies in the cognitive development of science. *Research policy*, 27(8), 807-821; Defazio, D., Lockett, A., & Wright, M. (2009). Funding incentives, collaborative dynamics and scientific productivity: Evidence from the EU framework program. *Research policy*, 38(2), 293-305; Morillo, F. (2016). Public-private interactions reflected through the funding acknowledgements. *Scientometrics*, 108(3), 1193-1204).

3. Research Funding

In my opinion, you need to provide a more thorough elaboration on your operationalization of joint funding and what it accounts for. On page 7 you state that funding implies collaboration “therefore the relationship between funders encompasses collaboration and funding.” However, on page 3 and 4 you stated that funders may be unaware that they are supporting the same project. As funding is the key concept of interest in your study, I think you need to elaborate more carefully on this concept and its operationalization. You may need to distinguish between funding for specific research projects and funding for individual research – e.g., grants for distinguished scholars. There may also be a difference whether a grant is developed and offered by two or more funders “by design” compared to different project grants that are combined by individual authors.

4. Data

I feel that the overall description of the data is rather short and raises many questions – some of these questions are crucial for your analysis. In particular

- The Cochrane Database of Systematic Reviews is a very particular type of publication and I am wondering whether the use of this data has an impact on the results.
- Cochrane reviews are updated when new evidence emerges. How does this affect the results?
- The link between publications and organizations is very important for your study. As organization names are often not standardized in bibliometric/publication databases, I wonder how you identified organizations in order to ensure that different name variants are counted as one organization.
- As you observe the data over a quite long period of time, you must explain how you deal with changes in organizational structures, e.g., mergers and acquisitions or spin-outs (particularly among firms). If you do not account for those events it is likely that report some artefacts instead of co-funding. Moreover, you would need to summarize the activities of subsidiaries of a single parent organization. An important example is your somewhat separation of the NIH and the National Cancer Institute which is part of the NIH. At least, you would have to explain your choice in great detail.
- How complete is the information on funders? Is this collected on a systematic basis by the data provider?

5. Methodological Aspects

One of the key findings of your study is that co-funding is linked to success in terms of impact measured by the g-index. However, even though you present “only” correlation analyses, I feel that there is something like an endogeneity problem – like in econometric studies- as you cannot observe the underlying quality of a project. Projects with higher quality are more likely to attract funding from different sources and attract – at the same time – more forward citations. Hence, you would either need more sophisticated econometric designs to assess the (causal) relationship between co-funding and citations or you would at least need to thoroughly discuss the endogeneity problem.

Further comments

6. Importance of geography

It found it a bit surprising that the presentation of your results starts with a geography of science story as this is nothing that I would have expected based on your literature review. However, the finding raises the question whether your literature review misses important reflections on the predominant organization of research funding, i.e., the importance of national funding with some exceptions such as funding provided by the European Union.

7. Hypothesis

In many disciplines, including the economics of science as a research area related to your study, hypotheses are developed based on the reviewed literature. Your hypothesis is fairly detached from the literature review and it is unclear how you develop this hypothesis. I recommend building hypotheses based on the literature. However, there is an additional issue with building hypotheses. As your analysis is predominantly descriptive, I am wondering what the benefit for formulating a single hypothesis is.

8. Structure

I encourage you to thoroughly review the structure of your arguments and the manuscript. It seems that some arguments are not well integrated and that you refer to results before clearly explaining your research design and your empirical strategy.

9. Implications

I am wondering what the implications of your work – based on a more thorough exploration of collaborative funding as suggested above – may be. Currently, the manuscript does not clearly elaborate on the implications for science managers, policy makers, and academics.

10. Forward and Backward Citations

It is not entirely clear to me which publications are included in the dataset and which are not included. Many literature streams distinguish among forward and backward citations (instead of incoming and outgoing citations).

Good luck with the development of your manuscript!

Reviewer

Review form: Reviewer 2

Is the manuscript scientifically sound in its present form?

Yes

Are the interpretations and conclusions justified by the results?

No

Is the language acceptable?

Yes

Do you have any ethical concerns with this paper?

No

Have you any concerns about statistical analyses in this paper?

No

Recommendation?

Major revision is needed (please make suggestions in comments)

Comments to the Author(s)

The manuscript 'The hidden influence of communities in collaborative funding of clinical science' by Vasan and West looks at the network structures that emerge, when funding agencies are connected if they support the same publication through their funding. The authors find that papers that were funded by more than one agency do better in terms of impact. Furthermore, the authors use a g-index to quantify success in funding, indicating that funders do better in terms of supporting high impact papers when they are co-funding research. As for funders that seek to fund with various dis-connected funders have higher success than being embedded in a network with strongly interconnected funders. As for the latter, the authors find a landscape of funding agencies that is very geographical.

The manuscript itself touches a very interesting subject. However, this reviewer is afraid that the results and their interpretation are a bit problematic. The observation that the funders network points to geographically bound clusters. Such an observation is putatively a consequence of the funding mechanisms that are mostly domestic, as the majority of grants go to domestic groups. Such limitations do not exist for companies that can co-operate and fund worldwide. As such, this reviewer is afraid that such a network in the way the authors constructed need to be taken with a grain of salt. The connections between international funding agencies are probably a

consequence of international collaborations that were domestically funded. In other words, it would be good to see what happens to the network, if papers that were published by purely national groups were discarded and only papers with international organizations were considered. Note that international collaborative papers are in the minority as most of the research output is a question of national efforts.

As the manuscript does not distinguish between international and purely national papers, it would be important to see if the success of papers with multiple grants and connections between funding agencies has an international component. In other words, it would be important to have a more granular analysis that figures if success is a question of domestic or international collaboration of funding agencies. With that said, the connections between funding agencies may also be question to distinguish if groups from different organizations worked together on a paper and coming with their own grants, or if the same team at one place had multiple grants. This reviewer is aware that this is a tall order, in that the underlying data may not be that granular to answer the question. At least, such possibilities should be discussed in the discussion section. As for conclusions the current manuscript makes interesting observations. However, the added value is a bit problematic. While connections between funding agencies are interesting from an academic point of view, policy implications are an issue, as funding agencies usually do not know which other agencies supported the same PIs or different ones, if the papers were written by more than one organization. As mentioned above the authors should discuss such points in more depth.

In Fig. 1 this reviewer is puzzled why the number of cited papers and journals drop and the number of citing journals and papers increases. Shouldn't that be the other way round, since papers that were published in 2019 had a much shorter time to receive citations than older papers.

Decision letter (RSOS-210072.R0)

Dear Mr Vasan

The Editors assigned to your paper RSOS-210072 "The hidden influence of communities in collaborative funding of clinical science" have now received comments from reviewers and would like you to revise the paper in accordance with the reviewer comments and any comments from the Editors. Please note this decision does not guarantee eventual acceptance.

Please submit your revised manuscript and required files (see below) no later than 21 days from today's (ie 09-Jun-2021) date. Note: the ScholarOne system will 'lock' if submission of the revision

is attempted 21 or more days after the deadline. If you do not think you will be able to meet this deadline please contact the editorial office immediately.

on behalf of Dr Mirco Musolesi (Associate Editor) and Marta Kwiatkowska (Subject Editor)
openscience@royalsociety.org

Associate Editor Comments to Author (Dr Mirco Musolesi):

Comments to the Author:

I would like to invite the authors to consider the comments of the reviewers and try to address them carefully. The reviews are very accurate and insightful: the Associate Editor believes that they will definitely help the authors to improve their manuscript.

Reviewer comments to Author:

Reviewer: 1

Comments to the Author(s)

Studying the influence of research funding is important to a wide and multidisciplinary academic audience but also for science managers and politicians. While I think that the overall research topic is suitable for the audience of Royal Society Open Science, I also think that the manuscript needs substantial improvements. Below I provide a more detailed discussion of some questions and comments that I have. I encourage you to thoroughly address the issues outlined below to improve your manuscript.

Main Comments

1. Research Question and Contribution

After reading your manuscript multiple times, I am still not sure what the research question is. It seems as if different versions of the research question come up throughout several sections of the manuscript, e.g. “how inter-connected are the agencies?” (page 1), “But how does collaboration between funders take place and how does it affect research success?” (page 2) “we quantify the importance and influence of collaborative funding in clinical science research from the perspective of funders” (page 4), or “how do the collaborative ties of a funder influence its success in funding?” (page 4). All these different versions indicate different foci of the paper and the analysis. I strongly recommend using one concise research question throughout the manuscript that is strongly linked to our empirical strategy. This will also support you in being clearer about your contribution.

2. Related Literature – Social Capital

You refer to the concept of social capital as a key reference point in the existing literature. However, I am not yet fully convinced that this concept can be applied in the context of your study. Burt and other authors refer to social capital as an individual level concept. However, you are essentially analyzing links among organizations. Hence, it needs to be thoroughly discussed whether and to what extent the concept of social capital can be applied on the organizational level. Moreover, I do not see the merits of this concept for guiding the reader through the paper and the analysis. The focus of the analysis does not seem to be on examining the role of social capital. Instead of focusing on social capital I recommend elaborating in much more detail on the related literature on research funding (e.g., Braun, D. (1998). The role of funding agencies in the cognitive development of science. *Research policy*, 27(8), 807-821; Defazio, D., Lockett, A., & Wright, M. (2009). Funding incentives, collaborative dynamics and scientific productivity: Evidence from the EU framework program. *Research policy*, 38(2), 293-305; Morillo, F. (2016). Public-private interactions reflected through the funding acknowledgements. *Scientometrics*, 108(3), 1193-1204).

3. Research Funding

In my opinion, you need to provide a more thorough elaboration on your operationalization of joint funding and what it accounts for. On page 7 you state that funding implies collaboration "therefore the relationship between funders encompasses collaboration and funding." However, on page 3 and 4 you stated that funders may be unaware that they are supporting the same project. As funding is the key concept of interest in your study, I think you need to elaborate more carefully on this concept and its operationalization. You may need to distinguish between funding for specific research projects and funding for individual research – e.g., grants for distinguished scholars. There may also be a difference whether a grant is developed and offered by two or more funders "by design" compared to different project grants that are combined by individual authors.

4. Data

I feel that the overall description of the data is rather short and raises many questions – some of these questions are crucial for your analysis. In particular

- The Cochrane Database of Systematic Reviews is a very particular type of publication and I am wondering whether the use of this data has an impact on the results.
- Cochrane reviews are updated when new evidence emerges. How does this affect the results?
- The link between publications and organizations is very important for your study. As organization names are often not standardized in bibliometric/publication databases, I wonder how you identified organizations in order to ensure that different name variants are counted as one organization.
- As you observe the data over a quite long period of time, you must explain how you deal with changes in organizational structures, e.g., mergers and acquisitions or spin-outs (particularly among firms). If you do not account for those events it is likely that report some artefacts instead of co-funding. Moreover, you would need to summarize the activities of subsidiaries of a single parent organization. An important example is your somewhat separation of the NIH and the National Cancer Institute which is part of the NIH. At least, you would have to explain your choice in great detail.
- How complete is the information on funders? Is this collected on a systematic basis by the data provider?

5. Methodological Aspects

One of the key findings of your study is that co-funding is linked to success in terms of impact measured by the g-index. However, even though you present "only" correlation analyses, I feel that there is something like an endogeneity problem – like in econometric studies- as you cannot observe the underlying quality of a project. Projects with higher quality are more likely to attract funding from different sources and attract – at the same time – more forward citations. Hence,

you would either need more sophisticated econometric designs to assess the (causal) relationship between co-funding and citations or you would at least need to thoroughly discuss the endogeneity problem.

Further comments

6. Importance of geography

It found it a bit surprising that the presentation of your results starts with a geography of science story as this is nothing that I would have expected based on your literature review. However, the finding raises the question whether your literature review misses important reflections on the predominant organization of research funding, i.e., the importance of national funding with some exceptions such as funding provided by the European Union.

7. Hypothesis

In many disciplines, including the economics of science as a research area related to your study, hypotheses are developed based on the reviewed literature. Your hypothesis is fairly detached from the literature review and it is unclear how you develop this hypothesis. I recommend building hypotheses based on the literature. However, there is an additional issue with building hypotheses. As your analysis is predominantly descriptive, I am wondering what the benefit for formulating a single hypothesis is.

8. Structure

I encourage you to thoroughly review the structure of your arguments and the manuscript. It seems that some arguments are not well integrated and that you refer to results before clearly explaining your research design and your empirical strategy.

9. Implications

I am wondering what the implications of your work – based on a more thorough exploration of collaborative funding as suggested above – may be. Currently, the manuscript does not clearly elaborate on the implications for science managers, policy makers, and academics.

10. Forward and Backward Citations

It is not entirely clear to me which publications are included in the dataset and which are not included. Many literature streams distinguish among forward and backward citations (instead of incoming and outgoing citations).

Good luck with the development of your manuscript!

Reviewer

Reviewer: 2

Comments to the Author(s)

The manuscript 'The hidden influence of communities in collaborative funding of clinical science' by Vasani and West looks at the network structures that emerge, when funding agencies are connected if they support the same publication through their funding. The authors find that papers that were funded by more than one agency do better in terms of impact. Furthermore, the authors use a g-index to quantify success in funding, indicating that funders do better in terms of supporting high impact papers when they are co-funding research. As for funders that seek to fund with various dis-connected funders have higher success than being embedded in a network with strongly interconnected funders. As for the latter, the authors find a landscape of funding agencies that is very geographical.

The manuscript itself touches a very interesting subject. However, this reviewer is afraid that the results and their interpretation are a bit problematic. The observation that the funders network points to geographically bound clusters. Such an observation is putatively a consequence of the

funding mechanisms that are mostly domestic, as the majority of grants go to domestic groups. Such limitations do not exist for companies that can co-operate and fund worldwide. As such, this reviewer is afraid that such a network in the way the authors constructed need to be taken with a grain of salt. The connections between international funding agencies are probably a consequence of international collaborations that were domestically funded. In other words, it would be good to see what happens to the network, if papers that were published by purely national groups were discarded and only papers with international organizations were considered. Note that international collaborative papers are in the minority as most of the research output is a question of national efforts.

As the manuscript does not distinguish between international and purely national papers, it would be important to see if the success of papers with multiple grants and connections between funding agencies has an international component. In other words, it would be important to have a more granular analysis that figures if success is a question of domestic or international collaboration of funding agencies. With that said, the connections between funding agencies may also be question to distinguish if groups from different organizations worked together on a paper and coming with their own grants, or if the same team at one place had multiple grants. This reviewer is aware that this is a tall order, in that the underlying data may not be that granular to answer the question. At least, such possibilities should be discussed in the discussion section. As for conclusions the current manuscript makes interesting observations. However, the added value is a bit problematic. While connections between funding agencies are interesting from an academic point of view, policy implications are an issue, as funding agencies usually do not know which other agencies supported the same PIs or different ones, if the papers were written by more than one organization. As mentioned above the authors should discuss such points in more depth.

In Fig. 1 this reviewer is puzzled why the number of cited papers and journals drop and the number of citing journals and papers increases. Shouldn't that be the other way round, since papers that were published in 2019 had a much shorter time to receive citations than older papers.

===PREPARING YOUR MANUSCRIPT===

If you have been asked to revise the written English in your submission as a condition of publication, you must do so, and you are expected to provide evidence that you have received

language editing support. The journal would prefer that you use a professional language editing service and provide a certificate of editing, but a signed letter from a colleague who is a native speaker of English is acceptable. Note the journal has arranged a number of discounts for authors using professional language editing services (<https://royalsociety.org/journals/authors/benefits/language-editing/>).

===PREPARING YOUR REVISION IN SCHOLARONE===

<https://royalsociety.org/journals/authors/author-guidelines/#supplementary-material> to

include a suitable title and informative caption. An example of appropriate titling and captioning may be found at https://figshare.com/articles/Table_S2_from_Is_there_a_trade-off_between_peak_performance_and_performance_breadth_across_temperatures_for_aerobic_sc_ope_in_teleost_fishes_/3843624.

Author's Response to Decision Letter for (RSOS-210072.R0)

See Appendix A.

Decision letter (RSOS-210072.R1)

Dear Mr Vasan,

It is a pleasure to accept your manuscript entitled "The hidden influence of communities in collaborative funding of clinical science" in its current form for publication in Royal Society Open Science. The comments from the Editors and reviewers are included at the foot of this letter.

Please ensure that you send to the editorial office an editable version of your accepted manuscript, and individual files for each figure and table included in your manuscript. You can send these in a zip folder if more convenient. Failure to provide these files may delay the processing of your proof.

Kind regards,
Royal Society Open Science Editorial Office
Royal Society Open Science

on behalf of Dr Mirco Musolesi (Associate Editor) and Marta Kwiatkowska (Subject Editor)
openscience@royalsociety.org

Associate Editor Comments to Author (Dr Mirco Musolesi):

The authors addressed all the concerns of the reviewers in a very convincing way. I believe that this is an extremely interesting paper, which might have an impact also outside the clinical sciences. I recommend this paper for acceptance.

Appendix A

Dr Mirco Musolesi
Associate Editor
Royal Society Open Science

Boston, July, 2021

Subject: Response letter to the editor

Dear Editor,

We enclose this response letter along with the revised submission of our article titled - *The hidden influence of communities in collaborative funding of clinical science*. We thank the editor for allowing us to resubmit and thank the reviewers for spending their valuable time reviewing our work and offering insightful feedback. Below, we provide summarized responses to the comments by the reviewers. The color **Orange** refers to the comments by the reviewers and color **Blue** shows our response.

Reviewer 1

Research Question and contribution. After reading your manuscript multiple times, I am still not sure what the research question is. It seems as if different versions of the research question come up throughout several sections of the manuscript

Thank you for seeking clarification and helping us refine our research questions and motivation. Our primary research question is the following : what is the role of collaborations between funders in generating high impact research? In the updated manuscript, we echo this research question based on two major subheadings - inference on collaborative funding and its implications on high impact research. This analysis is conducted through the following steps. First, we extract the different characteristics of funders such as the type of funder (government, private, nonprofit), primary country of operation (United States, United Kingdom) present within the network. This allows us to learn about patterns of collaborations, important questions such as, are regional collaborations more prevalent than international collaborations? Do we see an emergence of collaborations based on the type of organization? Next, we look at the implications of these network patterns on research impact. We measure the impact of funded research through a *g-index* metric that uses the citations received by the funded publications, which also indicates efficiency in funding clinically relevant research. In sum, these two steps allows us to investigate the potential value of collaborative funding on research output. We hope the new revisions provides additional clarity. Below we attach the relevant paragraph added to the revised manuscript:

The primary goal of this paper is to examine the role of collaborations between funders in generating high impact research and its implications in achieving higher research success from the perspective of funders. We conduct our research analysis in two broad steps. First, we map the connectivity of funders

through a co-funding network and explore the characteristics of communities created through these collaborations. Next, we measure the research value generated through these funder collaborations. In particular, we control for authorship and compare the value of solo funding versus co-funding in generating high impact research. Furthermore, we examine the network structures of funders that is associated with research success. This type of analysis may allow a funder to strategically position itself within this network. We also see this work useful for researchers trying to identify connected funders and for policy makers examining the research success of different funding model combinations. While the underlying data pertain to clinical science, we believe that the methods used in this paper to examine funding connectivity can be applied to other scientific domains that depend critically on collaborative funding such as climate change and public health.

Related Literature – Social Capital

You refer to the concept of social capital as a key reference point in the existing literature. However, I am not yet fully convinced that this is concept can be applied in in the context of your study. Burt and other authors refer to social capital as an individual level concept. However, you are essentially analyzing links among organizations. Hence, it needs to be thoroughly discussed whether and to what extent the concept of social capital can be applied on the organizational level. Moreover, I do not see the merits of this concept for guiding the reader through the paper and the analysis. The focus of the analysis does not seem to be on examining the role of social capital. Instead of focusing on social capital I recommend elaborating in much more detail on the related literature on research funding

Thank you for these comments. The reviewer is correct. We did not adequately address the issue of social capital. We have better focused the discussion around research funding and have added several references that support our claims, as the reviewer as suggested. In particular, we provide additional references to discuss the role of research funding in defining science policy. Here is the updated text from the manuscript :

Funders have a broad role in the development of science that includes the power to determine the next era of science research topics through rigorous grant proposal reviews [36] to picking the next researchers [9]. By virtue of providing financial support, funding agencies create the infrastructure required for scientists to pursue and conduct research, thus yielding a mediated and indirect influence on science [10]. While grant funding has minimal impact on the productivity of a researcher [27], it has been found that grant-sponsored projects get published in high impact journals [59] and therefore account for higher citations [61], a realization that persists across multiple domains [15, 55]. It has been discussed that this is partly because a researcher with a funded project typically has access to better resources and funds to travel and is usually more motivated to conduct research than otherwise. The differences between funding and research outputs can also be affected by the grant mechanisms distributed to investigators (e.g.NIH's R01, R21) [37]. Overall, funders create the organizational structure for authors to conduct research, making them important participants of science.

In addition, we provide more clarity about the applicability of social capital to our work. We discuss the previous work on inter-organizational networks in more depth to help operationalize its applications to funder networks. The text from the manuscript is stated as follows:

The ideas of pursuing social capital is motivated at an individual level, but it can also be useful at an organizational level, when thinking about resource dependence, market constraints, and partnerships [5]. The social capital embedded within the firm-university collaborations is found to be crucial in facilitating knowledge transfer [53] and for building a common view of goals[40]. Moreover, institutions can overcome organizational boundaries by building a diverse range of alliances [29] and sustaining relationships by seeking local partners[52]. This previous work highlights the crucial value of networks at an organization level. In terms of funders, the collaborative funding partnership scan serve as a bridge for projects that are resource scarce [54]. For our project,the social network of funders is examined through the lens of publications, where multiple funders may be acknowledged together for supporting a project. We use this network to examine collaborative funding and its relationship to research success.

Research Funding

In my opinion, you need to provide a more thorough elaboration on your operationalization of joint funding and what it accounts for. You may need to distinguish between funding for specific research projects and funding for individual research – e.g., grants for distinguished scholars. There may also be a difference whether a grant is developed and offered by two or more funders “by design” compared to different project grants that are combined by individual authors.

This is an important point and something we hope to address at some point in the near future. We fully agree that this is indeed a limitation of our study. Unfortunately, our data set does not provide this kind of resolution. It does not distinguish between the different types of joint funding connections. We consider both hidden and explicit funder connections because it is out of the scope of this paper to delineate between these two kinds of collaborations at the scale of our study. One possible way is to identify the grants associated with publications and check if the funders were listed on the same grant. The current data is limited in such mappings and would require a considerable effort at the scale of hundreds of thousands of papers. We are hoping that these data features are available at some point soon because these would be interesting questions to explore.

Furthermore, the "role" of the funder in the study (i.e. research design, resource sharing, monetary support) also affects the involvement of the funder and in translation their investment in their joint funders. Such granularity of associations is left for future work and we mention these problems in depth in the paper. We see this manuscript as an initial step towards mapping these funder connections and provide a layout for exploring these regional and international collaborations.

Data

I feel that the overall description of the data is rather short and raises many questions – some of these questions are crucial for your analysis. In particular..

The Cochrane Database of Systematic Reviews is a very particular type of publication and I am wondering whether the use of this data has an impact on the results.

The reviewer has highlighted an important element of the data. We use the Cochrane Database of Systematic Reviews as a way of focusing the study on clinically relevant research. This kind of research is a unique mixture of private and public investment. This allows for exploration of funding models and funding collaborations across these two major funding sources. Eventually, we would like to expand to all funders in all disciplines after we better understand the dynamics at this scale.

The Cochrane journal is a globally recognized journal that reviews clinical studies. The reviews present in the journal has been subject to several meta-analysis examining the quality of reviews presented (For example, Moja, Lorenzo P., et al. "Assessment of methodological quality of primary studies by systematic reviews: results of the metaquality cross sectional study." *Bmj* 330.7499 (2005): 1053.). Previous work has looked at the journal bias in the citations included in the paper (see Conduct and reporting of citation searching in cochrane systematic reviews: A cross-sectional study. *Research synthesis methods*11(2), 169–180 (2020)) and the search process of citations to include in the reviews (See Briscoe, S., Bethel, A., Rogers, M.: Conduct and reporting of citation searching in cochrane systematic reviews: A cross-sectional study. *Research synthesis methods*11(2), 169–180 (2020)). In addition, prior work has examined the applicability of the Cochrane reviews in clinical settings by looking at the publications that cite the reviews (See Shen, J., Li, Y., Clarke, M., Du, L., Wang, L., Zhong, D.: Production and citation of cochrane systematic reviews: a bibliometrics analysis. *Journal of Evidence-BasedMedicine* (2014))

This indicates that backward/forward citations from this journal are clinically relevant, not only in terms of the references that get chosen for review (backward citations) but also the use of the Cochrane reviews in creating new research (forward citations). The citations from these systematic reviews have influenced new science policies, for example, modifying citation bias (See Briscoe, S., Bethel, A., Rogers, M.: Conduct and reporting of citation searching in cochrane systematic reviews: A

cross-sectional study. *Research synthesis methods* 11(2), 169–180 (2020)). Indeed this classification using Cochrane journal is not an established resource, but we believe that this is a starting place with over 400k publications over three decades to examine the role of funders in clinical science.

Cochrane reviews are updated when new evidence emerges. How does this affect the results?

For this study, we do not examine the evidence or results presented within the publication. We simply look at the number of citations embedded within each Cochrane review publication. However, this is a pertinent point and we thank the reviewer for suggesting this. It could be that citations to an old, outdated review could be given undue credit for success. This is an excellent idea for a follow-up. It would require hand curating the positive/negative reviews and determining when evidence is updated, but this would be even more useful than looking just at citations. Given the challenge in doing this at scale, we chose to examine citations as a proxy for success but would eventually want to look at the citations with this kind of resolution.

The link between publications and organizations is very important for your study. As organization names are often not standardized in bibliometric/publication databases, I wonder how you identified organizations in order to ensure that different name variants are counted as one organization.

Thank you for bringing up this important point about disambiguation of organization names. This is a big problem in the bibliometric work, especially when it concerns author names. Our group has done significant work disambiguating institutional names just for own university collaborations. It is a lot of work but important for this kind of analysis. In the case of funder names we use the Global Research Identifiers Database (GRID, <https://grid.ac/>). The database provides a unique GRID id for each organization name, along with the type of institution, location. As of June 2021, the database is estimated to match 101k organizations from 219 countries. This feature is well-integrated with the Digital Science portfolio of companies. The Dimensions API is constantly curated and matched with the up-to-date identification of organizations. More information can be found at <https://www.dimensions.ai/blog/boost-your-research-organizational-data-using-grid-and-the-dimensions-api>.

As you observe the data over a quite long period of time, you must explain how you deal with changes in organizational structures, e.g., mergers and acquisitions or spin-outs (particularly among firms). If you do not account for those events it is likely that report some artefacts instead of co-funding. Moreover, you would need to summarize the activities of subsidiaries of a single parent organization. An important example is your somewhat separation of the NIH and the National Cancer Institute which is part of the NIH. At least, you would have to explain your choice in great detail.

The temporal nature of funding connections is an important point. The reviewer is correct to highlight this. We point out in the discussion section that the connections between funders may be a result of mergers and acquisitions, particularly in the pharmaceutical company. In a similar fashion, institutes and companies may dissolve and no longer exist, making the connection not useful for future collaborations. Such inter-organization dynamics are out-of-scope of our current work, but we discuss such implications on our analysis in the paper. We note this as a limitation of our study and agree 100% with the reviewer of this limitation. As this data improves, we hope that this is eventually available to researchers, but at this point, not even the largest and well resourced providers of this data (e.g., Clarivate, Scopus, and Dimensions, etc.) have this level of detail.

How complete is the information on funders? Is this collected on a systematic basis by the data provider?

The information about funders is extracted from the GRID database, which contains information about 101k countries from 219 countries. The distribution of disambiguated organizations by type of organization, primary country of operation can be found at <https://grid.ac/stats>. In addition to the statistics presented on the website we are unaware of any systemic biases involved in the data collection of the global organizations. Additional clarity on the coverage of organizations present in the database would add richness and confidence to our results, but at this moment this was the best

database we were able to find that contained curated organization names.

Methodological Aspects

One of the key findings of your study is that co-funding is linked to success in terms of impact measured by the g-index. However, even though you present “only” correlation analyses, I feel that there is something like an endogeneity problem – like in econometric studies- as you cannot observe the underlying quality of a project. Projects with higher quality are more likely to attract funding from different sources and attract – at the same time – more forward citations. Hence, you would either need more sophisticated econometric designs to assess the (causal) relationship between co-funding and citations or you would at least need to thoroughly discuss the endogeneity problem.

Thank you for highlighting the problem of endogenous variables impacting our findings. Throughout the paper, we try to stress that we are not claiming that co-funding will lead to higher success. We instead highlight the association between funding collaborations and research impact. The two way relationship between funding and citations is a noteworthy problem deeming further examination. Funders may choose to fund projects with good ideas and/or fund established investigators working on cutting edge ideas. On the other hand, funders may fund infrastructure needed to create high impact research. In our current research set up, we are unable to conduct interventions at the time of publication to test the value of specific funders or funding models. A particular line of questioning concerns measuring the changes in research success if the publication were not cofunded. This is an important question but through our results we are unable to make prescriptive claims about where to allocate funds and who to collaborate with.

With regards to confounding variables, we control for multi-authorship which is known to influence higher citations. This control allows us to investigate the rate of success between two publications in the categories of solo funding and co-funding. In the updated manuscript, we further clarify the other confounding factors, such as author reputation and journal prestige, that may affect our results. We acknowledge these exogenous variables in the updated version and present the limitations as ideas of further investigation.

Further Comments

Importance of Geography

It found it a bit surprising that the presentation of your results starts with a geography of science story as this is nothing that I would have expected based on your literature review. However, the finding raises the question whether your literature review misses important reflections on the predominant organization of research funding, i.e., the importance of national funding with some exceptions such as funding provided by the European Union.

Thanks for this suggestion. We should have done a better job incorporating the geographical element. We present a few citations in the manuscript that have found similar results on the role of geography in generating collaborations. In particular we present the following papers in the manuscript that examine at the role of geographical proximity in collaborations - Plotnikova, T., Rake, B.: Collaboration in pharmaceutical research: exploration of country-level determinants. *Scientometrics*. 98(2), 1173–1202 (2014); Bignami, F., Mattsson, P., Hoekman, J.: The importance of geographical distance to different types of rd collaboration in the pharmaceutical industry. *Industry and Innovation*27(5), 513–537 (2020).

The papers highlighted here and in the manuscript indirectly show the presence of such regional proximity in collaborative funding but do not directly test this hypothesis. The papers listed above focus on institutional collaboration and do not necessarily translate to the presence of geographic proximity in funder collaboration. Since funders may prefer to fund local teams, as a consequence of national policies or by choice, the appearance of communities of national funders show that authors tend to collaborate at a national level as well. In other words, our work provides even further evidence that exchange of ideas is mediated at the regional level, whether it be between authors, institutions,

or funders.

Hypothesis

In many disciplines, including the economics of science as a research area related to your study, hypotheses are developed based on the reviewed literature. Your hypothesis is fairly detached from the literature review and it is unclear how you develop this hypothesis. I recommend building hypotheses based on the literature. However, there is an additional issue with building hypotheses. As your analysis is predominantly descriptive, I am wondering what the benefit for formulating a single hypothesis is.

We thank the reviewer for this suggestion. Our analysis is predominantly descriptive, but we do have questions informed by the literature. We have modified the paper to make this clearer. Our primary hypothesis concerns testing the associated relationship between funder networks and funder success. This is a pertinent question because building funder partnerships is a tedious process, facing several impediments concerning governance structure, data management, and project timelines (See ref Lyall, C., Bruce, A., Marsden, W., Meagher, L.: The role of funding agencies increasing interdisciplinary knowledge. *Science and Public Policy*40(1), 62–71 (2013)). Indeed, building successful partnerships require good communication and a shared vision but is often difficult to create and sustain funder collaborations due to varied scientific varied particularly around allocation of funds (See, Lowe, P., Phillipson, J.: Barriers to research collaboration across disciplines: scientific paradigms and institutional practices. *Environment and Planning A*41(5),1171–1184 (2009)). These differences are further reinforced due to national and disciplinary boundaries, making certain funders distant. Indeed, funders have the possibility of easing the constraints of collaborations through severe policy changes and such actions can have a positive impact on science. However, the benefits of seeking distant partnerships is unclear. By creating an hypothesis oriented towards this, we can examine the association between network structure (i.e., collaborations) and funding success.

Structure

I encourage you to thoroughly review the structure of your arguments and the manuscript. It seems that some arguments are not well integrated and that you refer to results before clearly explaining your research design and your empirical strategy.

Thank you for the suggestion and providing us with the opportunity to improve the structure of the manuscript. We do this in the following steps. First, we provide a background of our work by explaining the benefits of collaborative funding in clinical research. This sets up the research questions, which examine the role of collaborations between funders in generating high impact publications. Next, we explain the reasoning behind selecting the Cochrane database as our set of core clinical science papers and stress the benefits in selecting this specific data source for our analysis. We believe this allows the reader to better interpret the results and recognize the limitations. After defining the research question and explaining the data, we state our network hypothesis concerning distant partnerships and research success.

We present the results in the following manner. First, we provide characteristics about the funder network, in particular, the communities of funder that organically emerge through publications. The updated results section allows the reader to better interpret the global knowledge spillover through funding. Next, we describe the findings from our network hypothesis and clarify the implications of our findings. In this part, we make it clear that the results do not follow causal claims but instead show correlated features between network structure and research success. Finally, in the discussion section, we further highlight the implications of our findings. In particular, we explain that our results allow researchers to identify connected funders and policy managers to make inferences about the global funding landscape and test the effect of different funding models. Our goal with these improvements is to better explain the different components of our research question and allow the reader to recognize the implications/ limitations of our work for science funding. We hope these improvements to the existing structure of the paper to improve readability and interpretation of the results.

Implications

I am wondering what the implications of your work – based on a more thorough exploration of collaborative funding as suggested above – may be. Currently, the manuscript does not clearly elaborate on the implications for science managers, policy makers, and academics.

Thank you for raising this point. Ultimately, we want this work to inform science managers, policy makers, and academics, and we did not do a good enough job of this in the original version of the manuscript. We have added an additional section that addresses the implications of the research. Below is the new paragraph we included:

We note that the drivers of success in collaborative funding could be confounded with other factors. It is possible that a research project receives funding from multiple funders due to the inherent quality of the research or the reputation of the authors. On the other hand, funders may be acknowledged for infrastructure investment. It is indeed difficult to conduct interventions to truly tease out the value of funders in a project and by extension the value of their collaborations. However, the patterns of connectivity between funders can be useful to both academics and policy makers. As we find in this work, funders regionally co-located are much more likely to share resources than otherwise. This is a useful finding for authors and academic investigators as they begin to build their own personal network of funders for their research. Evaluation of science policies is an important component of research funding. In this work we find that funders self-organize into national groups and funders that have higher success tend to occupy a central position collaborating with multiple disconnected funders. We view this work as a blueprint for policy makers and science managers to make inferences about funding policies that enable knowledge spillover across countries and examine the research output of such collaborations.

Forward and Backward Citations

It is not entirely clear to me which publications are included in the dataset and which are not included. Many literature streams distinguish among forward and backward citations (instead of incoming and outgoing citations).

In the paper we use the terminology incoming and outgoing citations which is synonymous to using the terms forward and backward citations. The entire dataset is used in the analysis (forward and backward cited papers). The only separation of the dataset is in examining the funder success. In this case, we filter papers that were cited by the Cochrane (backward citations). The paper text is updated to make this differentiation more explicit:

We then extract the incoming citations (citing papers; papers that cite the Cochrane journal or forward citations) and outgoing citations (cited papers; papers that are cited by the Cochrane journal or backward citations).

Reviewer 2

Research Funding

The observation that the funders network points to geographically bound clusters. Such an observation is putatively a consequence of the funding mechanisms that are mostly domestic, as the majority of grants go to domestic groups. Such limitations do not exist for companies that can co-operate and fund worldwide. As such, this reviewer is afraid that such a network in the way the authors constructed need to be taken w a grain of salt.

Thank you for your comment regarding the inference of the funding network. We agree with the reviewer that the agreements of grants for national entities (e.g. NSF, European Union) may have limitations in funding or collaborating with international partners. This is not a major limitation in the case of pharmaceutical companies that may have subsidiaries present in multiple countries and

may have an easier time finding partners. This is indeed reflected in the funder network with several groups of national funders and a distinct large group of pharmaceutical companies. In the discussion component of the paper, we provide possible explanations of our findings, including the varied priorities of funding of government and pharmaceutical companies. In addition, we would like to highlight that the goal of this work is not to compare and contrast the funding mechanisms of different funders but to map the global funding partnership created through publications and its effect on citations. We see this network for funders as a map that examines clinically relevant funding collaborations.

The connections between international funding agencies are probably a consequence of international collaborations that were domestically funded. In other words, it would be good to see what happens to the network, if papers that were published by purely national groups were discarded and only papers with international organizations were considered. Note that international collaborative papers are in the minority as most of the research output is a question of national efforts.

We note the limitation of the study in differentiating between the known versus unknown funder collaborations and the effect of authorship in creating links between funders. This level of granularity is unfortunately unavailable to us at the moment and would incur tedious human effort to classify the papers. We leave this component of the paper for future work that hopefully can be done. At a broader level, we believe that the funder network does a good job of explaining the regional and global policies that may create new partnerships. We view this funder network as another layer of complexity in explaining the role of funders and authors in creating domestic and international collaborations.

As the manuscript does not distinguish between international and purely national papers, it would be important to see if the success of papers with multiple grants and connections between funding agencies has an international component. In other words, it would be important to have a more granular analysis that figures if success is a question of domestic or international collaboration of funding agencies.

Thank you for your this comment on differentiating national and international collaborations. This comment sparked a new avenue of questions that we did not think of in the earlier draft. We now explore the effect of geographic borders in creating international collaborations. See below the paragraph we have included regarding this:

Overall, we find that funders are more likely to collaborate with others in the same country than outside the country. Of all the papers that had multiple funders, 39517 (73.5%) publications had funders from the same country while only 14221 (26.4%) publications featured funders from multiple countries. Indeed this observation could be attributed to the funding policies from national entities that focus on regional collaborations thus creating interdisciplinary boundaries between countries [51]. However, if such restrictions didn't exist, how would the collaborative funding landscape look? Based on random simulations, we find that if funders pursued collaborations with funders irrespective of their country, 57.70% of the co-funded papers would feature international collaborations. This indicates that the real-world network has much fewer international partner-ships than we would expect at random. In other words, the country of operation highly influences the ability for two funders to collaboratively fund a project. Our findings offers further confirmation to the hypothesis of localized science, where resource sharing and exchange of ideas appears to be country mediated

With that said, the connections between funding agencies may also be question to distinguish if groups from different organizations worked together on a paper and coming with their own grants, or if the same team at one place had multiple grants. This reviewer is aware that this is a tall order, in that the underlying data may not be that granular to answer the question. At least, such possibilities should be discussed in the discussion section.

We thank the reviewer for mentioning the difference between hidden and known collaborations. We recognize this as a limitation of the study in aptly differentiating the types of connectivity between funders. When referring to funder collaborations we refer to both types of collaborations, unless noted explicitly. We include both types because it is out of the scope of this paper to delineate between

these two kinds of collaborations at the scale of our study. One possible way is to identify the grants associated with publications and check if the funders were listed on the same grant. The current data is limited in such mappings and would require hand labeling hundreds of thousands of papers. This differentiation will be an important follow-up to this study.

Implications

As for conclusions the current manuscript makes interesting observations. However, the added value is a bit problematic. While connections between funding agencies are interesting from an academic point of view, policy implications are an issue, as funding agencies usually do not know which other agencies supported the same PIs or different ones, if the papers were written by more than one organization. As mentioned above the authors should discuss such points in more depth.

Thank you for highlighting the communication between inference and policy implications. This paper is meant to provide an overview of the funder connections. We agree that the funder network calls for interesting observations on the science policy that has created several domestic and international collaboration but may lack causal features since funders may be unaware of the connections. This is an added limitation of this study. We mention the funder network therefore to encompass hidden and explicit funder connections and control for multi-authorship as a source of confounder. In response to this comment, we add the following to the revised submission:

We note that the drivers of success in collaborative funding could be confounded with other factors. It is possible that a research project receives funding from multiple funders due to the inherent quality of the research or the reputation of the authors. On the other hand, funders may be acknowledged for infrastructure investment. It is indeed difficult to conduct interventions to truly tease out the value of funders in a project and by extension the value of their collaborations. However, the patterns of connectivity between funders can be useful to both academics and policy makers. As we find in this work, funders regionally co-located are much more likely to share resources than otherwise. This is a useful finding for authors and academic investigators as they begin to build their own personal network of funders for their research. Evaluation of science policies is an important component of research funding. In this work we find that funders self-organize into national groups and funders that have higher success tend to occupy a central position collaborating with multiple disconnected funders. We view this work as a blueprint for policy makers and science managers to make inferences about funding policies that enable knowledge spillover across countries and examine the research output of such collaborations.

Further Comments

In Fig. 1 this reviewer is puzzled why the number of cited papers and journals drop and the number of citing journals and papers increases. Shouldn't that be the other way round, since papers that were published in 2019 had a much shorter time to receive citations than older papers

Thank you for seeking clarification on our terminology. Cited papers refers to the papers that are cited by the Cochrane journal (backward citations) and citing papers refers to the papers that cite the Cochrane journal (forward citations). The number of citing papers does not refer to the total citations received by Cochrane papers but indicates the list of papers that were published in that year that cited Cochrane. Indeed, as time goes on more Cochrane papers exist and therefore more papers start to cite other Cochrane papers. We have made the text more explicit to indicate this difference:

We then extract the incoming citations (citing papers; papers that cite the Cochrane journal or forward citations) and outgoing citations (cited papers; papers that are cited by the Cochrane journal or backward citations).

We thank the reviewers for providing us constructive feedback. The comments have greatly improved the manuscript. We also thank the editor for allowing us to revisit and improve the manuscript based

on this feedback.

Kind regards,
Kishore Vasani